# Morphological evolution of bird wings follows a mechanical sensitivity gradient determined by the aerodynamics of flapping flight

Jonathan A. Rader [1] ✉ & Tyson L. Hedrick [1]

The physical principles that govern the function of biological structures also mediate their evolution, but the evolutionary drivers of morphological traits within complex structures can be difficult to predict. Here, we use morphological traits measured from 1096 3-dimensional bird wing scans from 178 species to test the interaction of two frameworks for relating morphology to evolution. We examine whether the evolutionary rate ($\sigma^2$) and mode is dominated by the modular organization of the wing into handwing and armwing regions, and/or the relationship between trait morphology and functional output (i.e. mechanical sensitivity, driven here by flapping flight aerodynamics). Our results support discretization of the armwing and handwing as morphological modules, but morphological disparity and $\sigma^2$ varied continuously with the mechanical sensitivity gradient and were not modular. Thus, mechanical sensitivity should be considered an independent and fundamental driver of evolutionary dynamics in biomechanical traits, distinct from morphological modularity.

Form−function relationships are one of the pillars of biodiversity. Morphological features have diverged in size and shape among lineages and impart different abilities to interact with the environment and compete for finite resources[1–4]. The evolution of individual morphological traits is not always directly and solely linked to their biomechanical function, though. Individual traits within a biomechanical system can contribute in varying degrees to the functional output of the whole. Traits working together to perform some function are the hallmark of morphological integration[5–10]. "Function" in this context can refer to the physical output of force to accomplish a variety of tasks including (but not limited to) feeding, locomotion, sexual display, or competition, as well as a variety of other behavioral and potentially physiological attributes.

Modularity exists when function differs between clusters of integrated traits[5–10]. The degree of integration among these coevolving traits and their organization into mosaics of semi-independent modules are mediated by their shared development and by the magnitude of their impact on functional output[5–8,11–13]. Locomotor modules[14] are a subset of functional modularity wherein regionalization of biomechanical function (i.e., running vs. flying) leads to the development of morphological (and potentially physiological) traits specific to those tasks. As natural selection acts to shape the functional output of biomechanical systems, each of the individual traits may experience selective pressure commensurate with their relative contribution to the output of the system[10,11,13,15]. Therefore, in a biomechanical context, the strength of the relationships between morphological traits and their mechanical function (termed mechanical sensitivity) may be an important driver of their evolutionary dynamics (i.e. tempo and mode[16–18]).

The first description of the relationship between mechanical sensitivity and evolutionary dynamics focused on four-bar linkage systems[16], particularly in the jaws of teleost fish[19,20] and the raptorial appendages of mantis shrimp[21]. Here, each link can be thought of as a discrete morphological module. The modules with the greatest impact

[1]Dept. of Biology, University of North Carolina at Chapel Hill, Chapel Hill, NC, USA. ✉e-mail: jrader@email.unc.edu

on the transmission of force or motion in a four-bar linkage system also have the greatest mechanical sensitivity[16], which correlates with a shift in evolutionary mode (from Brownian motion toward Ornstein-Uhlenbeck) and to a higher evolutionary tempo[17,18]. However, despite finding a similar coupling of mechanical sensitivity and evolutionary dynamics in the four-bar linkage systems in two disparate taxa, the generalizability of these results remains hampered by a lack of comparable studies of other morphological traits in biomechanical systems beyond the four-bar linkage[17]. As such, studying morphological modularity and evolution in systems with different biophysical interactions, such as the fluid-structure interactions in flight or swimming, can fill some of the missing picture of the patterns and processes that shape the evolution of complex biological structures. We investigated how the evolutionary dynamics of wing shape in birds have responded to the interplay between the mechanics of aerodynamic force production in flight and morphological modularity within the wing.

Birds are diverse in their ecology and behavior, which manifests as differences in their flight style, morphology (see Fig. 1a), and performance. Bird wings must produce lift to support body weight during flight and asymmetrical forces for maneuvering. They must function at cruising speed, at low airspeeds during landing and maneuvering flight, and during supra-normal efforts for pursuit or escape flight. Furthermore, bird wings may experience trade-offs and constraints imposed by their structure and evolutionary development.

The geometry of a wing influences how it interacts with the air, and thus the lift and drag forces that it generates[22]. Consequently, wing shape in birds is related to flight and migration behavior[23-28] and numerous other aspects of avian biology[29-33]. Much work has been done describing how planform wing shape (2-dimensional shape in the wing span vs. chord dimensions) is related to avian aerodynamics[34-39]. However, wings are not two-dimensional structures. Three-dimensional (3D) shape attributes such as wing camber (the upward curvature of the wing's surface, see Fig. 1) contribute to aerodynamic forces produced by the wing[22,40,41], and the distribution of mass along the wing span impacts the cost of flapping[42] and maneuverability[43,44]. Because 3D attributes of the wing are tied to its function, they are also potentially evolutionarily labile and tunable features, worthy of consideration in the story of avian wing evolution[26,41]. Bird wings are also not static structures. The shape -both 2D and 3D- changes throughout the wingbeat cycle and is modified by birds, termed wing morphing, to accomplish various flight tasks[45-52].

Though the wing feathers create a generally contiguous wing surface, the avian wing is composed of multiple anatomical subunits[53]. The most obvious of these are associated with the major skeletal regions of the forelimb (Fig. 1b). The portion of the wing associated with the radius and ulna (and to a lesser degree, the humerus) is the armwing (AW), and includes the bony elements, muscles, tendons, and the secondary and tertial portions of the feathered wing surface[53]. The AW also supports the propatagium on its leading edge. The handwing (HW) is comprised by the bones of the wrist and hand as well as the primary portion of the feathered wing surface, but with minimal contribution from muscles and tendons[53]. The two wing regions (HW and AW) may be under differential selective pressures, or subject to different selective or developmental tradeoffs and constraints leading to regionalization of biomechanical function *sensu*[14] leading to evolutionary or morphological modularity within the wing[12,53]. For these reasons, we hypothesized that morphological modularity exists in the wing, dividing it into discrete armwing and handwing modules.

Wing moment of inertia for bending or rotation about the shoulder and aerodynamic forces from flapping flight both vary as functions of the distance from the base of the wing toward the tip. If wings are uniform in chord length and density along their span, aerodynamic forces and inertial moment increase as the square of this distance, producing the greatest magnitudes at the tip of the wing[54]. The Weis-Fogh[54] model is based upon two-dimensional strips taken

spanwise from the wing, but three-dimensional and fluid-dynamics effects (such as vortex formation) not described by this simplified model also exist. For instance, wings in gliding flight also experience a reduction in aerodynamic force production near the wing tip due to formation of tip vortices. However, in flapping wings, tip vortices interact with and may even enhance the leading-edge vortex, a primary source of lift in flapping wings, thus potentially strengthening the base-to-tip aerodynamic force gradient[55]. The magnitude of and balance between these inertial and aerodynamic effects are modulated in actual bird wings by root-to-tip tapering of both wing chord length and mass. The net result of flapping, irrespective of the influence of tip vortices and wing taper, is a general increase in the force per unit area[55] and inertial moment; a distal section of wing will experience greater aerodynamic force and greater inertial moment per unit area than a comparable section more proximal on the wing (see Supplementary Note 1 for more details). In turn, this means that a change in morphology in distal wing sections will yield a greater relative change in the forces experienced there than would result from a similar, more proximal, change in morphology. We therefore posit that the gradients of aerodynamic forces and inertial moment lead to a similar gradient of increasing mechanical sensitivity along the length of the wing, smoothly crossing the hypothesized junction of handwing and armwing modules (Fig. 1; Supplementary Note 1). If mechanical sensitivity is tied to the evolution of wing shape as it is in four-bar linkages[18], the gradient of mechanical sensitivity along the wing should result in a corresponding gradient of evolutionary dynamics of shape traits. Though wing shape changes dynamically through flight, the range of shapes that a wing can achieve via wing morphing is constrained by its static form. We investigated the relationship between mechanical sensitivity and the evolution of static wing morphology.

We identified two idealized patterns that might characterize how evolutionary dynamics in the wing will respond to the interaction of the mechanical sensitivity gradient with morphological modularity in the wing: (1) Evolutionary dynamics could follow the base to tip gradient in mechanical sensitivity established by flapping flight, irrespective of morphological modularity, producing a smooth root-to-tip pattern of increasing evolutionary tempo (Fig. 1c). (2) Alternatively, if evolution acts upon modules within the wing, the mechanical sensitivity gradient superimposed across the HW and AW modules would cause the HW region to have a distinctly faster evolutionary tempo than the AW, also with a possible shift to a different evolutionary mode across the wrist joint. In this second case, evolutionary tempo would vary less within these modules than among them (Fig. 1c), with a well-defined step up in evolutionary rate from the AW to the HW. Under both hypothesized evolutionary regimes, shape traits would display greater interspecific variation (termed morphological disparity) near the tip of the wing, with evolutionary tempo increasing from the base of the wing toward its tip. The primary difference is that in Hypothesis 2, both evolutionary tempo and morphological disparity should show a regionally discontinuous pattern between the HW and AW, separated by the wrist. Alternatively, if flapping flight has little influence on the evolutionary dynamics of wing morphology, we would not expect any particular root-to-tip gradient of evolutionary rate or morphological disparity. Instead, wing evolution might be shaped by tradeoffs among biomechanical, life history, and ecological traits, leading to unpredictable evolutionary patterns. We investigated these hypotheses using 3D surface scans of wings from 178 species representing 15 major lineages of birds (Fig. 1a), providing a basis for exploring regionalization and modularity of avian wing morphology and evolution.

Here, we demonstrate that the mechanical sensitivity concept can be applied to biomechanical systems other than four-bar linkages where it was first described[16]. Four-bar linkages transmit forces directly via interacting lever arms, whereas aerodynamic forces arise from a fluid-structure interaction. Despite the physical differences between

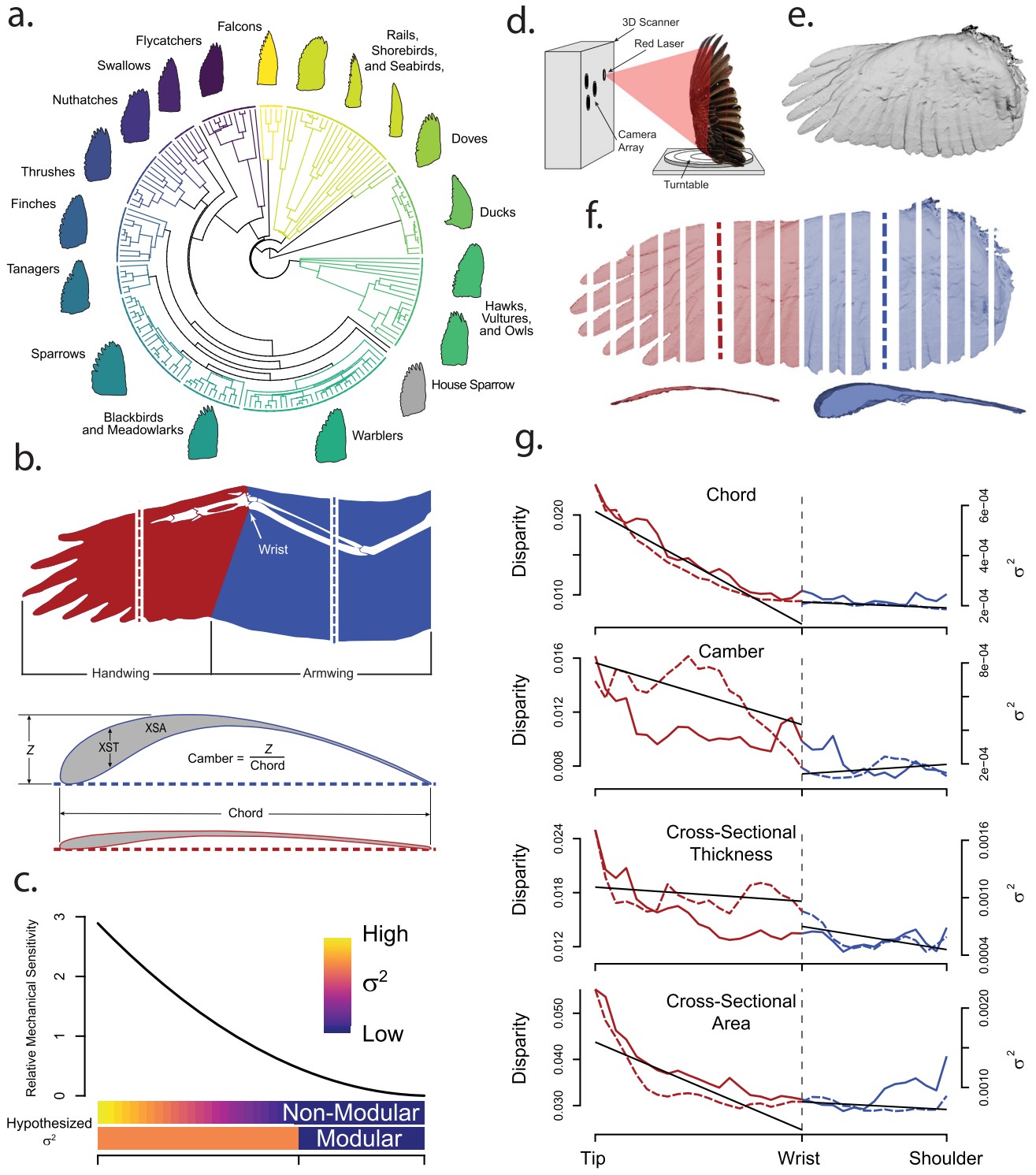

**Fig. 1 | Phylogenetic sampling, morphological traits, 3D scanning methodology, and evolutionary dynamics of bird wings.** Bird wings are complex biological structures with phylogenetically structured morphological variability **a** and are composed of musculoskeletal and integumentary elements **b**. Flight feathers form the aerodynamic surface of the wing, which is divided into two regions separated by the wrist, the handwing and the armwing **b**. Aerodynamic forces and inertial moment increase as the square of the distance from the base of the wing during flapping flight[54], creating an alternative model for wing mechanical sensitivity **c**. The evolutionary signature might follow the mechanical sensitivity gradient, or might be discretized by interacting with modularity in the wing **c**, hypothesized $\sigma^2$).

We used a laser scanner **d** to capture surface scans **e** of 1096 wings from 178 species of birds **a**. The tree was pruned from the Jetz et al. supertree[73]. We divided the wing into chord-wise slices **f** along the span, and measured chord, camber, cross-sectional thickness (XST), and cross-sectional area (XSA) from each slice (see **b**). **g** Morphological disparity (dashed line) and evolutionary rate ($\sigma^2$, solid line) for all these shape traits were greater in the handwing (red) than in the armwing (blue), and especially so near the wingtip. Regression discontinuity analyses (RDA) of morphological disparity (solid black lines) showed significant discontinuity across the wrist joint (see Table 1). The distinction across the wrist was less clear for $\sigma^2$, except in wing chord and marginally in XST.

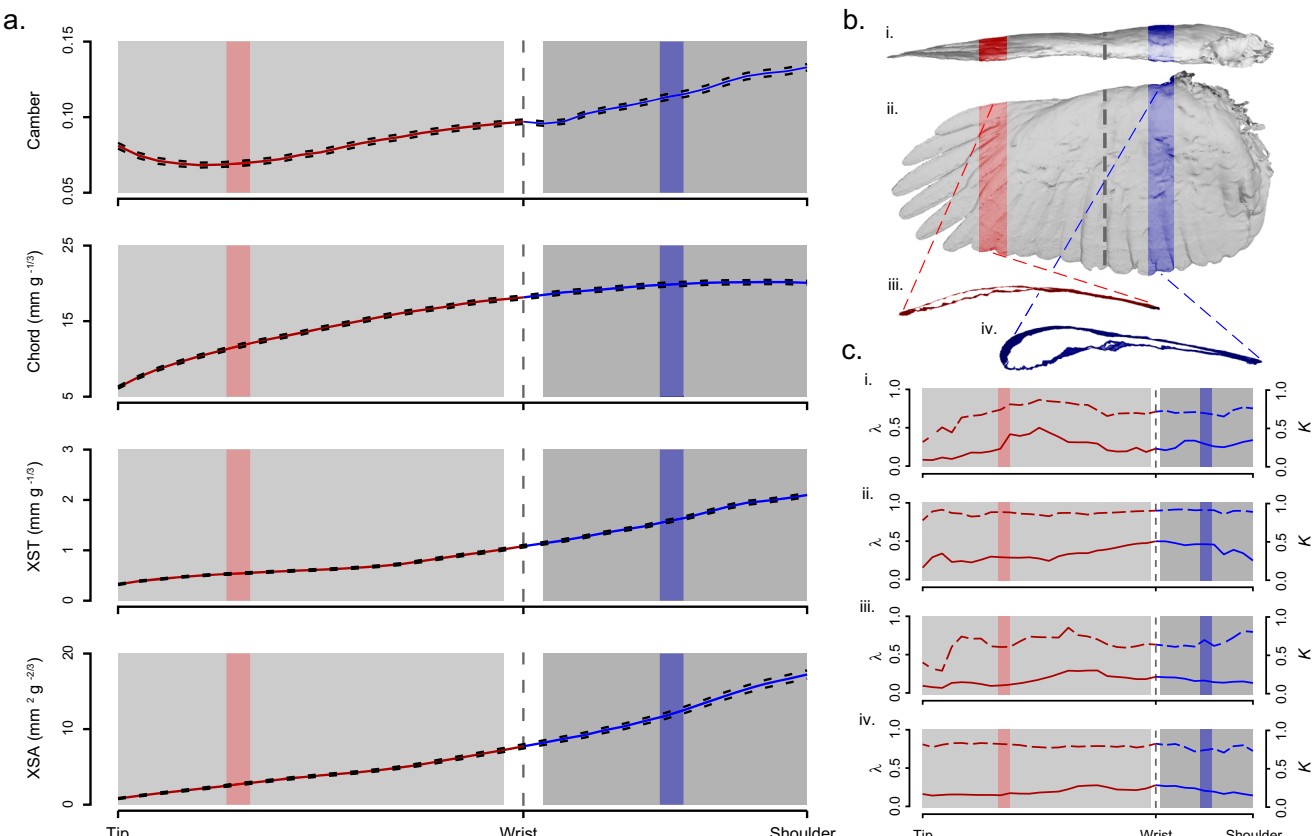

**Fig. 2 | Wing shape traits scaled by body size showed marked consistency across taxa, were of greatest magnitude in the armwing (dark gray region), and decreased away from the shoulder, across the wrist joint (vertical dashed line) and through the armwing (lighter gray region). a** Mean values (solid red/blue lines) ± SE (black dashed lines), across 1096 wings from 178 taxa, of camber, chord, cross-sectional thickness (XST) and cross-sectional area (XSA) in the handwing (red) and armwing (blue). The shaded gray boxes show the regions that were included in subsequent analyses of morphological and evolutionary modularity, and the red and blue shaded regions correspond with wing slices pulled from the scans (shown in **b**). The wrist was excluded. **b** Example of a 3D scanned wing from a Cooper's hawk (*Accipter cooperii*) in (i) frontal and (ii) planform views, with (iii) representative slices shown for the handwing (red) and armwing (blue). **c** Phylogenetic signal was high in all shape traits in both wing regions. Dashed red/blue lines show Blomberg's *K* for camber (i), chord (ii), XST (iii), and XSA (IV) along the length of the wing, and solid lines show Pagel's *λ*.

these systems, a common theme has emerged: morphological evolution is greatest when mechanical sensitivity is high. This applies to modular systems with discrete differences in sensitivity, but also along continuous gradients. Our results therefore suggest that mechanical sensitivity provides a general framework for the study of morphological and biomechanical evolution.

## Results

We scanned 1096 wings representing 178 species of birds with an average sample size of 6 individuals per species. Median wing camber across all slices, averaged within each species, ranged from 0.061 to 0.169, with an overall mean of 0.105. Armwing (AW) camber was greater than that in the handwing (HW, median ± MAD: 0.11 ± 0.021 vs. 0.072 ± 0.017). Mean chord ranged from 30.6 mm to 249.9 mm, with an overall mean of 64.3 mm. Median chord (scaled by dividing by $M_b^{1/3}$) was greater in the AW ($20.43 \pm 2.59$ mm g$^{-1/3}$) than in the HW ($13.16 \pm 2.53$ mm g$^{-1/3}$). Wing thickness at the most proximal measured slice of the AW varied among the study species from 0.41 mm to 4.38 mm g$^{-1/3}$ (mean = 2.10 mm g$^{-1/3}$) and tapered to the wrist joint. Thickness of the wrist joint ranged from 0.44 mm g$^{-1/3}$ to 2.36 mm g$^{-1/3}$ with an average of 1.14 mm g$^{-1/3}$. Wing thickness tapered further toward the most distal measured slice of the HW (range = 0.20 – 1.90 mm g$^{-1/3}$, mean = 0.50 mm g$^{-1/3}$). Wing cross-sectional area showed a similar pattern, tapering from a mean of 17.22 mm² g$^{-2/3}$ at the most proximal measured wing slice (range = 2.20 to 42.08 mm² g$^{-2/3}$) to a mean of

2.09 mm² g$^{-2/3}$ at the most distal measured slice (range = 0.64 to 19.70 mm² g$^{-2/3}$). The profiles of camber, chord, cross-sectional thickness (XST) and cross-sectional area (XSA) across the measured portion of the wing are shown in Fig. 2.

### Morphological modularity

The covariance ratio (CR) test[56] identified significant morphological modularity (CR < 1.0, see Fig. 3) in the log-transformation of all shape traits (camber CR = 0.79, p < 0.001; chord CR = 0.79, p < 0.001; XST CR = 0.87, p < 0.001; XSA CR = 0.87, p < 0.001), suggesting that the AW and HW are morphologically discrete subunits of the wing. Log-transforming the data removed the biasing effect of differing means between the regions[57], but a similar outcome was obtained from the raw data as well. Additionally, this result was robust to inclusion of the wrist in either the hand or arm region. The wing shape traits that we measured showed increasing trends from the tip of the wing toward its base (see Fig. 2). The result of this is that the values of each shape trait for a given slice are inherently more similar to closely situated slices than they are to more distant ones. We tested whether the existence of an underlying trend in the shape data would bias us toward finding a signal of modularity using the CR method by simulating wing shape traits with no phylogenetic structure and measuring CR in the simulated data (see Supplemental Fig. S5). We found that the simple existence of a trend in the data, linear or otherwise, is insufficient to produce a signal of modularity.

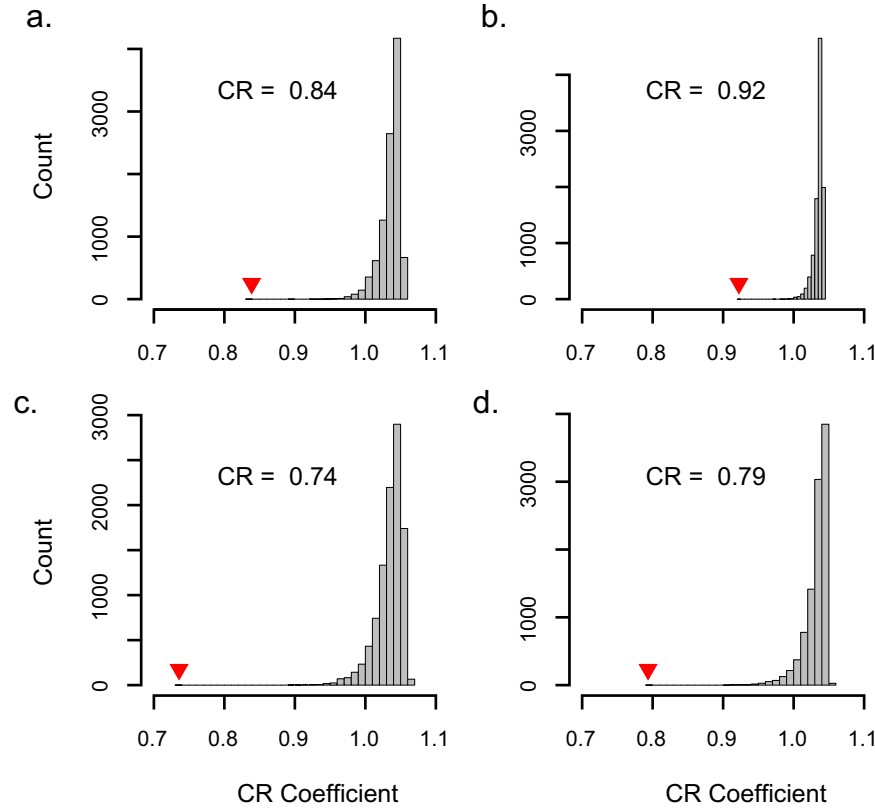

**Fig. 3 | Results of the covariance ratio test.** The covariance ratio test (CR)[56] identified significant modularity between the handwing and the armwing for **a** camber, **b** chord, **c** cross-sectional thickness, and **d** cross-sectional area.

**Table 1 | Phylogenetic signal (both Pagel's λ and Blomberg's K), morphological disparity, evolutionary tempo ($\sigma^2$) and regression discontinuity analyses for four wing shape traits: wing camber, chord, cross-sectional thickness (XST), and cross-sectional area (XSA)**

| Shape Trait | Mean Phylogenetic Signal (HW λ,K \| AW λ,K) | Mean Disparity (HW \| AW) | Mean $\sigma^2$ (HW \| AW) | Disparity RDA effect $p$ | $\sigma^2$ RDA effect $p$ |
|---|---|---|---|---|---|
| Camber | 0.69, 0.25 \| 0.71, 0.28 | 0.014 \| 0.008 | 0.0005 \| 0.0003 | 0.000156 | 0.721920 |
| Chord | 0.86, 0.32 \| 0.90, 0.41 | 0.012 \| 0.008 | 0.0004 \| 0.0002 | 0.000114 | 0.000351 |
| XST | 0.64, 0.17 \| 0.67, 0.16 | 0.018 \| 0.013 | 0.0009 \| 0.0006 | 0.0257 | 0.0282 |
| XSA | 0.79, 0.19 \| 0.76, 0.21 | 0.035 \| 0.030 | 0.0012 \| 0.0009 | 0.0141 | 0.487 |

## Morphological disparity

Disparity of camber was greater in the HW (mean = 0.014, see Table 1) than in the AW (mean = 0.008), with maximum (0.017) near the middle of the HW and a sharp downward transition through the wrist joint (Fig. 1g). Regression-discontinuity analysis (RDA) confirmed that the spanwise trend in camber is discontinuous about the wrist ($p < 0.001$). Disparity of chord exhibited no spatial discontinuity, had comparatively low values throughout the AW (mean = 0.007), and increased distal to the wrist. Mean chord disparity in the HW was 0.012, and the distinction between the HW and the AW was supported by RDA ($p < 0.001$). Disparity of XST and XSA shared similar patterns: disparity was greatest near the tip of the wing, and decayed toward the middle of the HW (consistent with the pattern depicted in Fig. 1c "ii"). In both cases, RDA showed marginal support for the discontinuity across the wrist joint ($p_t = 0.025$, $p_A = 0.014$).

## Evolutionary tempo and phylogenetic signal

For all shape traits, model-fit criteria (AICc) supported an Ornstein-Uhlenbeck (OU) model of evolution across all wing slices (all ΔAICc > 4). Evolutionary tempo ($\sigma^2$) among the shape traits (camber, chord, XST, and XSA) broadly showed trends similar to morphological disparity.

Greater evolutionary rates corresponded with higher disparity (Table 1). However, the discontinuity across the wrist joint was less distinct, except in chord (RDA $p < 0.001$). Discontinuity results were marginal for XST ($p = 0.028$) and non-significant for XSA and camber. Phylogenetic signal was high throughout the wing in all shape traits (all $K < 0.35$ and all $\lambda > 0.60$, see Table 1).

We found that the gradient patterns of both $\sigma^2$ and disparity were consistent across different partitions of the data (see supplementary Figures S3 and S4). Both passerine and non-passerine birds showed similar patterns of $\sigma^2$ and disparity, though the estimated values differed between the groups (see supplementary Figure S3). Also, a general gradient pattern is maintained through the rarefaction analysis (supplemental Figure S4), though random removal of taxa affects the estimates of $\sigma^2$ and disparity. Finally, we also found that our results were robust to different phylogenetic hypotheses. The gradient pattern of $\sigma^2$ was replicated across 1000 posterior draws of the avian phylogeny (supplemental Figure S2).

## Discussion

We tested whether modularity in bird wings has interacted with a gradient of biomechanical force production along the length of the

wing to shape morphological evolution. Our tests of morphological modularity confirmed that the HW and AW are discrete morphological modules within the wing, however we found little evidence that this modularity has influenced morphological evolution. Instead, we find that the physics of flapping is a dominant evolutionary driver. We hypothesized that a root-to-tip increasing gradient of mechanical sensitivity (the strength of the relationship between trait morphology and functional output[16,17]) arising from gradients of aerodynamic forces and inertial moment for flapping wings[54] (see Fig. 1c; Supplementary Note 1) would result in a matching gradient in evolutionary dynamics (especially evolutionary tempo).

Morphological disparity was significantly greater in the HW for all traits, and $\sigma^2$ largely followed a similar trend. Both measures decreased away from the wingtip even within the handwing, following the gradient of mechanical sensitivity for flapping flight but breaking the strictly modular expectation for evolutionary dynamics. There was significant linear regression discontinuity for $\sigma^2$ in wing chord and marginally for XST. However, $\sigma^2$ for camber and cross-sectional area were not discretely separated at the wrist, despite greater values near the wingtip. We found significant discontinuities in morphological disparity across the wrist joint in all traits, supporting discretization of the HW and AW as two morphological modules. Our results match our prediction that the HW and AW are morphological modules, but more consistent with our first hypothesis: evolutionary dynamics follow the mechanical sensitivity of the morphological traits. Specifically, we show that 1) evolutionary rate can track mechanical sensitivity within and not just among modules; thus the link between mechanical sensitivity and evolutionary rate does not depend upon morphological modularity and 2) evolutionary dynamics can track continuous mechanical sensitivity gradients without significant interaction with underlying morphological modules. We provide detailed discussion on wing modularity and its interaction with mechanical sensitivity in the following sections.

## Bird wings are modular structures

Gatesy and Dial[14] proposed that locomotor modularity (the integration of anatomical subunits, such as the hindlimbs or forelimbs, into functional subunits during locomotion) is responsible for the evolutionary diversification of avian morphology and locomotion, and potentially for the origin of flight. Here, we present a refinement to their view of modularity within the structure of the wing. Our results show that bird wings are complex structures composed of at least two morphological modules, the handwing (HW) and the armwing (AW), delineated by the wrist joint. We measured morphological disparity, a quantification of the occupancy of multivariate space such as that formed by multiple morphological axes (in this case, four axes: camber, chord, XST and XSA), along the wing. We found that morphological disparity is greatest in the HW, and especially so at the wingtip. We used Regression Discontinuity Analysis (RDA) to demonstrate that the patterns of morphological disparity were different in the HW and the AW (see Fig. 1g), providing evidence that the HW and AW are discrete modules.

In this study, we describe a test case where morphological modularity exists, but apparently without the strong functional disparity corresponding with the morphological modules. Hence, we show that morphological modularity is not necessarily tied to evolutionary modularity, but that mechanical sensitivity is a better predictor of evolutionary dynamics than modularity. Prior studies in other systems that have linked morphological and evolutionary modularity found that strong regionalization of biomechanical function, whatever the particular function may be, leads to strong regionalization of morphology. Morphological modularity has been documented in mammalian backbones, where a gradient of selective pressures along the length of the spine leads to regionalization of both form and function[15,58]. The flight feathers of bird wings form a set of serially-

homologous elements akin to vertebrae in the mammalian backbone, but experience a gradient of forces[54,59] rather than discrete regionalization of function. We note, though, that the relationship between form and function among the wing modules we describe herein is only implied, not experimentally validated for the variety of taxa considered here, and based upon a simple model of the distribution of per-area force and inertial moment in wings[54]. More detailed, complex and potentially accurate models of lineage-specific wing shapes might prove a fruitful path for further investigation into the evolution of bird wing shape but potentially at the cost of the generalizability of our present results.

Our finding that the HW and AW form discrete morphological modules does not imply that additional modularity cannot be found within wings. The skeletal, muscular, and integumental components of the wing might experience unique evolutionary pressures and tradeoffs that shape their evolution across multiple levels of organization[12,60]. For example, the thickness of the wing skeleton is tied both to its aerodynamics—thicker wings present more frontal area to the wind and produce more drag, and to its structural rigidity—thicker wing bones are more able to resist bending[61]. The geometry of the wing's feathered surface influences, in part, the magnitude and distribution of the forces the skeleton must withstand[54,62]. Therefore, these features might show morphological integration, i.e., that their morphologies coevolve within discrete regions of the wing, and the strength of the integration may vary along the length of the wing. Conversely, the flight feathers are serially homologous features arranged along the wingspan, whose relative sizes determine the dimensions and geometry of the wing surface[24,63]. Each of the flight feathers is potentially exposed to different evolutionary pressures based on its position on the wing[54], raising the possibility of further regionalization and modularity may exist among the flight feathers, with the extreme case being that each flight feather could be its own morphological and evolutionary module. Our 3D scanned wings did not permit investigation of modularity at these levels of organization, so it possible that a greater magnitude of modularity exists that could be uncovered by different measurement techniques.

## Trait evolution follows a gradient rather than modules

We predicted and found that bird wings show strong morphological modularity between the HW and the AW. We also predicted that evolutionary dynamics (tempo and mode) would differ significantly between morphological modules. While we did find that mean values of $\sigma^2$ were greater in the HW for all traits, evidence for evolutionary modularity was equivocal. Regression discontinuity analysis identified significant transitions in $\sigma^2$ for chord and marginally for XST, but not for camber or XSA. Instead, $\sigma^2$ for XST is consistent across much of the wing, with a notable increase near the wingtip. Camber $\sigma^2$ also shows an increase at the wingtip but is otherwise consistent within the HW and greater than in the AW. Though discontinuous across the wrist, $\sigma^2$ of all traits more closely approximated the gradient predicted by our first hypothesis. The discontinuities represent an inflection point in the function, rather than a step. We therefore found little support for evolutionary modularity within the wing and propose that evolutionary change of the shape traits discussed here is not beholden to morphological modules, but instead follows a smooth gradient along the span of the wing.

The relative aerodynamic force and inertial moment experienced by each unit area of the wing increase in a root-to-tip gradient[54,55]. The result of this is that altering the shape of distal regions of the wing will yield a greater change in the force/inertia regime of that region than would be caused by a similar proportion shape change more proximally. The outcome of these biomechanical gradients is a span-wise increase in mechanical sensitivity toward the tip of the wing. Mechanical sensitivity influences evolutionary dynamics of morphological traits, biasing toward higher rates of

evolutionary diversification when mechanical sensitivity is high[17,18], as we found in the HW. Greater morphological disparity and faster evolutionary tempo in the HW (and particularly its distal tip region) relative to elsewhere along the span support our hypothesis that selective pressures driving morphological evolution in avian wings are related to the distribution of aerodynamic and inertial forces along the span of the wing. Our results are also consistent with prior work demonstrating that 2D, planform wing shape has diverged primarily near the wingtips[24], and demonstrate that 3D shape traits (camber, XST, and XSA) behave similarly.

We predicted that evolutionary dynamics of wing morphology follow a gradient of mechanical sensitivity increasing from the base of the wing toward its tip. Based on simple aerodynamic and inertial models, we predicted that the general shape of the gradient would resemble an exponential curve (see supplemental material for details). However, we caution that we have made no explicit predictions as to the magnitude of that increase, or what might constitute a "significant" gradient. Because different bird taxa rely on gliding and flapping flight to varying degrees[28], we expect that the sensitivity gradient itself might be a feature that varies similarly. For example, we predict that the sensitivity gradient strongest for hummingbirds, which rely almost solely on flapping[64–67] and weakest for specialized gliders like albatrosses and condors[68–71], with other lineages falling somewhere on a spectrum between those end points. Our finding that the values of $\sigma^2$ and disparity differ between passerine and non-passerine birds reinforces our assertion that the sensitivity gradient might vary among taxa. Our intent was to assess whether the large representation of passerines in our sample biased us toward finding a gradient of evolutionary dynamics if passerines follow that pattern, but other birds do not, and not to test for differences in the magnitude of the gradient among taxa, so we hesitate to interpret that result further.

Prior studies linking mechanical sensitivity to evolutionary dynamics in 4-bar linkage systems have documented transitions in evolutionary mode (i.e. Ornstein-Uhlenbeck vs. Brownian motion) in addition to a shift toward higher rates[17,18]. However, we found no shift in mode across the wrist joint. The OU model was best supported by AICc in all shape traits across the entirety of the wing. This is unsurprising for camber and XSA, as the RDA models for these traits did not highlight significant transitions in evolutionary rate ($\sigma^2$) across the wrist joint. However, there were significant differences in $\sigma^2$ between the wing regions for chord and, marginally, for XST, but without a shift in evolutionary mode. We posit that the lack of sharp transitions in evolutionary dynamics at the wrist joint, despite trait disparity analysis supporting discretization of the HW and AW into separate morphological modules, stems from a continuous gradient of increasing mechanical sensitivity along the span of the wing.

## Additional considerations

We investigated the evolution of static wing shape, but bird wings are dynamic structures. Planform wing shape is dynamically and deliberately modified by birds in flight, termed "wing morphing" to modify aerodynamic performance[45,46] and to react to transient perturbations[47–49]. Three-dimensional shape traits like camber and span-wise twist vary as the wing cycles through its flight stroke and when acted upon by aerodynamic forces in flight[50], and the range of motion at the wing joints is a stronger predictor of flight style than 2D wing shape[51]. The static shape of a wing influences the breadth of different morphologies that the wing can adopt via morphing[51] and is likely predictive of the envelope of aerodynamic performance that a morphing wing can produce[49,52]. As yet, though, a systematic understanding of how static wing shape affects the manner and to what degree birds can dynamically alter the shape of their wings in flight remains elusive and is an area of active investigation[49–51]. Our present results provide an important step forward in understanding how the biomechanics of avian flight drives the evolution of their wings, but

more importantly, the mechanical sensitivity framework that we applied here to describe the evolution of static wing shape may also provide a valuable roadmap for future exploration of morphing wings.

Several shape indices have been developed to facilitate the broad taxonomic sampling necessary to explore how wing shape in birds relates to their behavior and ecology. The most widely adopted of these is the handwing index (HWI)[24,63], which serves as a proxy for wing aspect ratio. The present results suggest that the wingtip is especially evolutionarily labile relative to the rest of the wing and shape traits there are likely to be tuned to the various flight and lifestyle pressures among avian taxa. Wingtip shape indices such as HWI can capture variation in the handwing, but the utility of wingtip indices remains limited. HWI provides an imperfect proxy for wing aspect ratio. The proportion of the AW varies among avian taxa (from approximately 30–60% of wing length in our sample). Birds with identical HWI can have very different AR. Furthermore, AR, by itself, misses several other traits that directly influence various aspects of flight performance. Coefficient of lift and lift to drag ratio are both functions of an interaction between wing camber and AR[22,40,41], but camber is not captured by any wingtip shape index. Additionally, wing mass (and as a proxy, wing volume) affects the inertial moment of the wing, influencing the energetic cost of flapping and the ability to use wing inertia for maneuvering. We therefore suggest that while wingtip shape indices can be valuable proxies in broad views of ecomorphological relationships (e.g.[30,31]), they provide too coarse of a morphological measurement to be useful in taxonomically-broad studies of flight biomechanics.

We assembled a dataset of 3-dimensional wing shape in a broad taxonomic sample of birds. Our analyses show that the wing is divided into at least two morphological modules separated by the wrist, the handwing and the armwing, and that shape divergence was greatest in the handwing. We tested competing hypotheses of how evolutionary dynamics act upon the wing modules, and found that morphological disparity was significantly modular within the wings, but that evolutionary tempo followed a gradient of mechanical sensitivity along the span of the wing that was predicted by a blade element model of flapping flight aerodynamics and inertial moment[54]. This expands our understanding of evolutionary dynamics of complex biological structures, demonstrating that morphology can be tuned along continuous gradients in addition to previously described modular processes[17,18]. Our results concur with prior observations that mechanical sensitivity drives evolution of biomechanical traits. We also demonstrate that the linkage between mechanical sensitivity and evolutionary dynamics is not specific to four-bar linkages, but also exists in other systems. The mechanical sensitivity of morphological traits in biomechanical systems may therefore be fundamental to the evolution of form and function.

## Methods

### Wing scanning and measurement

We collected three-dimensional wing shape data from spread wings in the collection at the North Carolina Museum of Natural Sciences (NCMNS) in Raleigh, NC using a NextEngine 3D Scanner Ultra HD laser scanner (NextEngine, Inc., Santa Monica, CA; Fig. 1d). Sample sizes of each taxon were limited by the availability of specimens in the NCMNS collection, but when available, we scanned 16 individuals per species, maintaining a balanced sex ratio. Scan resolution was set to optimize scanning time while preserving surface detail. Resolutions ranged from 78 dots per cm$^2$ for large wings to 6300 dots per cm$^2$ for smaller wings. The scanned wings (Fig. 1e) were processed using a MATLAB program (MATLAB r2014b, The MathWorks, Natick, MA, USA) that extracted the vertices from the 3D object files, creating point clouds in the shape of the wings. We aligned the wing point clouds to a common coordinate system with the X-axis extending along the length of the wing from base to tip, the Y-axis

along the chord from leading to trailing edge, and Z through the thickness. Wing length ($r$) was measured as $X_{max}$ - $X_{min}$, and wingspan as $2r$. Because the wings were removed from the body during preservation, we were unable to account for the width of the body in our measure of wingspan.

Three-dimensional shape traits were measured by subdividing the wings into chord-wise slices along their span (Fig. 1f). To facilitate direct comparisons between wings and among taxa, we set the width of the slices to be 1/25th of the distance from the wrist joint to the tip of the wing, ensuring that all wings would have the same number of handwing (HW) slices. The number of slices representing the armwing (AW) was allowed to vary, as the proportion of HW vs. AW differs among taxa. To standardize subsequent analyses, we restricted the dataset to 35 span-wise slices (25 HW and 10 AW), which reflects the number of slices in the taxon with the shortest AW. In addition to standardizing the analyses, substantial trauma occurs during the removal of the wing during preservation, so excluding the proximal AW slices also reduces the influence of preservational artifacts.

We measured four shape traits from each wing slice, focusing on attributes of the wing that are expected from first principles to directly influence the aerodynamic forces and inertial moment of the wing. We favored this approach over a geometric morphometric (GM) approach because, while GM may provide higher resolution shape information, the link between form and function in a GM framework is less direct. Chord was measured as $Y_{max}$ - $Y_{min}$, and cross-sectional area (XSA) was measured as the area contained within a spline fitted to the perimeter of the wing section in the $Y/Z$ plane. The maximum distance in the Z-dimension (i.e., the greatest distance from the upper wing surface to the lower wing surface) was recorded as the maximum cross-sectional thickness (XST). Camber was calculated as ($Z_{max}$ - $Z_{min}$) / ($Y_{max}$ - $Y_{min}$).

Body mass ($M_b$) was recorded from museum tag data where available. When mass was not available from specimen tags, a species mean value was filled in from the CRC Handbook of Avian Body Masses[72]. Measurements of wing length, area, chord, and thickness were scaled by dividing by body mass taken to the appropriate power ($M_b^{1/3}$ for linear measures and $M_b^{2/3}$ for areas) and summarized within each taxon. Subsequent analyses were conducted on species median values for each wing slice.

### Phylogenetics
Phylogenetic analyses were based upon the Jetz. et. al. [73] super tree from Birdtree.org[74], which includes 10,000 Bayesian posterior draws of the tree. The tree was pruned to include only taxa in our scanned wing dataset. Handling of the tree, data, and phylogenetic analyses was done using tools from the *Phytools*[75] and *geiger*[76] packages in the R Statistical Computing Environment version 4.1.0[77]. We calculated two metrics of phylogenetic signal: Blomberg's $K$[78] and Pagel's $\lambda$[79] for each wing slice using the 'phylosig' function in the *Phytools* package.

### Morphological modularity analysis
**Modularity.** To assess whether the HW and AW are morphologically distinct modules, we used the covariance ratio (CR) proposed by Adams[56]. This test compares the covariance among traits within a putative module to covariance among the modules. The test statistic (CR) ranges from 0 to positive infinity, with values between 0 and 1 representing greater covariance within putative modules than among them, signaling morphological modularity. CR greater than 1 indicates morphological integration, and a lack of modularity[56]. This test was implemented using code provided in the supplement of Adams' description of the method[56]. Because the mean values of the shape traits differ between wing regions, in addition to conducting the modularity test on the isometrically scaled data, we used a $log_{10}$-transformation to mitigate any biasing effect from the difference in

means. Furthermore, because the wrist joint affects wing camber, thickness, and XSA, its effect on the modularity analysis was difficult to predict. To test whether inclusion of the wrist influenced our modularity interpretations, we iterated our analysis, including the wrist in each the HW and the AW regions while excluding it from the other. We also removed the wrist from consideration entirely, and only analyzed regions proximal and distal to it. Because inclusion of the wrist had no impact on the modularity analysis, and for the sake of simplicity, we present results excluding the wrist since this avoids arbitrarily assigning it to either the AW or HW region.

**Disparity.** Morphological disparity is a measure of the variation in traits among taxa. We compared morphological disparity in each of our shape traits for each wing slice using the 'dispRity' function in the *dispRity* package[80] in R. We compared disparity between the HW and the AW using regression discontinuity analysis (RDA) using the 'rdd_reg_lm' function in the *rdd_tools* package[81] in R. Regression discontinuity analysis is a statistical tool to assess changes in slope or intercept in a temporal or spatial trend across an assigned $X$-axis cutoff point, in our case, the wrist joint. A difference in disparity in the same shape traits between different regions of the wing would indicate a difference in evolutionary lability as well.

### Evolutionary modularity analysis
**Evolutionary tempo and mode.** To test whether different morphological modules expressed different evolutionary dynamics, we fit Brownian motion (BM), Ornstein-Uhlenbeck (OU), and early-burst (EB) evolutionary models to each of the shape traits at each of the wing slices using the 'fitContinuous' function in the *geiger* package in R. We used Akaike's Information Criterion corrected for small sample size (AICc) to determine the most suitable model for each trait and estimated evolutionary rate ($\sigma^2$) for each wing slice using that model. We tested for differences in patterns of $\sigma^2$ between the wing regions using regression discontinuity analysis (RDA) as described above.

**Effects of sampling and phylogenetic uncertainty.** Passerines account for 113 of our 178 species taxonomic sample of birds. Though the Passeriformes is a large and morphologically diverse lineage, its strong representation in our sample could bias our main findings if the lineage differs systematically from other birds. We took two approaches to determine if evolutionary dynamics of wing morphology in passerines differ from other birds and thus influence the broader interpretation of our results. First, we divided our dataset into passerines and non-passerines, and applied the $\sigma^2$ and morphological disparity analyses as described above to each of those subsets. Second, we conducted a rarefaction analysis in which we removed taxa and iterated the $\sigma^2$ and disparity analyses on the remaining taxa. In this analysis, we randomly removed 1 to 172 (of a total 178) taxa, meaning that we conducted the $\sigma^2$ and disparity analyses on sets of 6 to 177 species; the removed taxa were replaced before subsequent subsampling. Further details of these analyses are presented in the supplementary material.

A phylogenetic tree is a hypothesis of the relatedness among species. The analysis of $\sigma^2$ can be influenced by the assumed tree topology (branching structure and branch lengths). To assess the effect of tree topology on our estimates of $\sigma^2$, we iterated our analysis across 1000 posterior draws of the bird tree with varying topologies. We then took the median $\sigma^2$ of the output at each wing slice along with the median absolute deviation (MAD) to summarize their central tendency and variance respectively (see supplemental material for details). Disparity, unlike $\sigma^2$, is insensitive to tree topology.

### Reporting summary
Further information on research design is available in the Nature Portfolio Reporting Summary linked to this article.

## Data availability

All shape trait data and the phylogenetic tree generated in this study have been deposited, without restrictions, in figshare using https://doi.org/10.6084/m9.figshare.16899580.

## Code availability

All analysis scripts are available at https://doi.org/10.6084/m9.figshare.16899580.

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

## Acknowledgements

We thank the North Carolina Museum of Natural Sciences, especially Brian O'Shea and John Gerwin, for access to specimens and assistance throughout the wing scanning operation. We also thank Alaowei Amanah, Eli Bradley, Sophia Chizhikova, Elliot Cho, Lucy Herrero, Russel Lo, Raghu Padma, Alva Rönn, Colton Sanders, Sarah Yaghoubi, and Zhitong Yu, for their tireless efforts scanning wings. Pranav Khandelwal provided thoughtful discussion and feedback throughout the project. Lindsay Waldrop, Pranav Khandelwal, Brenna Hansen and Sonja Friman gave helpful feedback on an earlier draft of the manuscript. We are grateful to the Associate Editor, Stephanie McClelland and two anonymous reviewers for their helpful commentary and suggestions that improved this manuscript. Funding for this work was provided by the National Science Foundation IOS–1253276 to TLH, NSF DEB 1737752 to Daniel R. Matute, and a Sigma Xi GIAR award to JAR. We also thank NextEngine, Inc. for providing financial support for wing scanning equipment and software.

## Author contributions

Conceptualization: J.A.R., T.L.H., Methodology: J.A.R., T.L.H., Investigation: J.A.R., T.L.H., Visualization: J.A.R., Supervision: T.L.H., Writing—original draft: J.A.R., Writing—review & editing: J.A.R., T.L.H.

## Competing interests

The authors declare no competing interests.
