## [Peer Review File · Nature Communications]

Reviewers' Comments:

Reviewer #1:

Remarks to the Author:

General comments

The authors aim to test an interesting set of hypotheses about the drivers of wing shape in birds by assembling a very large sample of 3D scanned wings from a range of taxa. Although the questions are interesting, there are a significant number of unclear aspects of the analysis that undermine the results as presented.

What is the null hypothesis (lines 138-158)? The authors describe two possible patterns and the expected predictions from those, but what would no pattern (or a different pattern) be evidence for? How do you determine the gradient? How steep does a gradient have to be to show evidence for mechanical sensitivity of wing shape? What determines if there is a cutoff or discontinuity?

Broadly, there should be much much more description of the methods either in the text or in SI. As written, it would not be possible to replicate the methods, as you will see by the number of questions and clarifications in the methods section. By the time I reached the results, I was totally at a loss about how the authors carried out the CR analysis.

In the end, the authors might be correct in their conclusions, but it not currently possible to determine from the information presented.

Specific comments

Introduction

- Lines 63-66 are a little confusing in the way that "function" is used three times with seemingly different connotations.
- Hanging a lot of intellectual framework on the Felice et al paper. Is each link really independent (lines 74-75)? Can the handwing really exist without the armwing? Are they actually functionally separable, given that each is involved in the production of aerodynamic forces? I have similar questions about the Felice et al. argument.
- Wings aren't 2d -- agreed. But wings are also not static in 3d. The authors certainly know this, but it represents an important caveat to this work overall. What looks morphologically similar in 3d (or 2d) might move dissimilarly in the dynamics of flight (and vice versa).
- Line 115, I think that the Gatesy & Dial citation is not appropriate here. Those authors were talking about locomotor modules not in the GM "modules" sense, but rather as evolutionarily decoupled complexes (forelimb, hind limb, tail -- not within wing functional separation).

Methods

- Details of the PCA are not at all clear from the text (lines 165-174). It seems like the authors are doing a geometric morphometric analysis, but the usual discussion of those methods is missing (centering, scaling, etc.). Is this GM analysis? What steps did the authors use? What software, etc.? In GM, care must be taken to make sure that landmarks are homologous to one another or that there are sufficiently dense sliding or semi-sliding landmarks that homology can be approximated (it seems like this would be necessary for the CR analysis). Do the authors test whether upsampling large specimens or downsampling small specimens had an effect on the analysis? I would be concerned that different numbers of points per specimen might cause problems (or maybe this just reflects not understanding the approach).
- Lines 171-172, Are the PC axes descriptions of the loadings? It is very unclear how these axes were determined.

- Lines 175-178, how does this method alter the AW vs. HW dichotomy when there are differing numbers of segments in each (i.e., where the wrist falls relative to the segments)? Did the authors try a per-AW or per-HW approach?

- Can the authors be certain that using only 2/3 of the AW does not bias the results? Are any whole-bird scans available for validation?

- Lines 192-193 -- using the species medians is probably not necessary. Phylogenetic comparative methods can utilize multiple observations per specimen. One of the Garland & Ives (or Ives & Garland) papers discusses the use of measurement error, which is statistically indistinguishable from having multiple specimens per species.

- Phylogenetic tree -- What was the source of the branch lengths and how were they scaled (if at all) when collapsing branches during pruning? As a phylogenetic tree is a hypothesis of relative relationships are there alternative phylogenetic hypotheses (topology or branch lengths)? How sensitive is the analysis to topology and/or branch length scaling?

- I have some concerns about taxon sampling. Although passerines represent half of avian taxonomic diversity, morphologically they are rather homogenous. My gestalt feeling is that there is much more diversity in wing shape outside of passerines than within it. I wonder if some kind of rarefaction analysis could show that the authors are achieving "saturation" in wing disparity with the current sampling.

- Lines 220-228 -- The authors need to include more information about the RDA analysis. What models were tested? How were they compared? It appears from Fig 1g that only linear discontinuity was tested (?vs. a linear model with no discontinuity?). This seems at odds with the theoretical prediction in Fig 1c of some kind of exponential decay in relative mechanical sensitivity (which seems like it might be an appropriate fit to the observed data). Also, maybe it is my right-handed bias, but I find it counterintuitive to have the axes run from tip to shoulder. I would have expected the reverse (which also might mean that a regular polynomial will fit well to these data).

Results

- Lines 258-260 -- It's not clear how the CR analysis can be used on camber, chord, etc. The method was designed for multivariate shape data based on landmarks. Here it seems to be applied to scalar data. The fact that randomized data are not centered on CR = 1, suggests that something is amiss with this analysis.

Reviewer #2:

Remarks to the Author:

The authors present a study on the modularity, mechanical sensitivity and evolution of wing shape across birds. Their main goal is to try and determine whether wing shape evolution (represented by disparity and evolutionary rates) is more driven by modularity in the wing, or mechanical sensitivity to wing loading and aerodynamics. In general I like this study a great deal and am happy to see some of these concepts applied to a new system. I only have one somewhat major critique and a few minor points for clarification.

My only "major" critique is with the modularity side of the analysis. I am not convinced that the modularity signal they find between the AW and HW sections isn't actually just a methodological artefact. The CR method, if I have this right, essentially asks whether the trait values within modules are more close to each other than they are to trait values between modules. In this study, the traits are taken from a series of wing slices that run in series along the long axis of the wing. The two modules hypothesized in this study are also aligned next to each other in the same manner. It seems to be that it would be hard for these two to not be seen as separate modules. Unless the wing has a very strange shape, neighboring slices are always going to be more similar to each other relative to non-neighboring ones. Due to this, all the slices in HW, for example, are

more close to each other, than the slices in AW with the exception for maybe the 1-2 near the boundary, but those are likely not enough to shift the overall pattern. In short, I believe the modularity signal being seen is due to comparing a series of neighboring features, which will be more similar to each other, against non-neighboring ones, making me question whether there really are two distinct modules here.

Some more minor clarifications:

Pg. 8, Ln 188-190: Does using the average mass for a bird species when the specimen mass is not available create a problem if the specimen in question ends up being an outlier in terms of size?

Pg. 10, Ln 220-228: One aspect of the methods that I am fuzzy on is how some of these analyses are done when there are differing numbers of slices within the AW module. For instance, when looking at disparity of the trait values, it says that disparity is measured for each slice. But if there are not the same number of slices in the AW region across species, how do you determine which ones are used for a specific disparity measure? For instance, if one bird has 10 slices and another has 8, how are those 8 slices mapped onto the ten for measuring disparity? And do you have 8 measures of disparity or 10?

The discussion opens with several paragraphs which feel like a rehash of the introduction. Some of this can be streamlined/cut.

REVIEWER COMMENTS

Reviewer #1 (Remarks to the Author):

General comments

The authors aim to test an interesting set of hypotheses about the drivers of wing shape in birds by assembling a very large sample of 3D scanned wings from a range of taxa. Although the questions are interesting, there are a significant number of unclear aspects of the analysis that undermine the results as presented.

Response: We thank the referee for the positive evaluation of our manuscript and will attempt to address the concerns that have been raised in the comments below.

What is the null hypothesis (lines 138-158)? The authors describe two possible patterns and the expected predictions from those, but what would no pattern (or a different pattern) be evidence for?

Response: The absence of a pattern would suggest that the physics of flapping flight does not impose a gradient of mechanical sensitivity along the wing, or that morphological evolution in the wing does not respond to mechanical sensitivity in a way that resembles the previously-described relationship in 4-bar linkage systems. Other patterns might hint at influences from biomechanics in the wing, life history, ecology, or other factors. Predicting those patterns and their possible causes *a priori* is difficult. We have added presentation of our null hypothesis to the introduction (Lines 168 - 171):

“Alternatively, if flapping flight has little influence on the evolutionary dynamics of wing morphology, we would not expect any particular root-to-tip gradient of evolutionary rate or morphological disparity. Instead, wing evolution might be shaped by tradeoffs among biomechanical, life history, and ecological traits, leading to unpredictable evolutionary patterns.”

How do you determine the gradient?

Response: Our hypothesis of the gradient is borne out of the physical gradient of aerodynamic forces created by the velocity gradient along flapping wings (see the section entitled “Flapping” in the supplemental material for a derivation). We have clarified this in the manuscript (Line 138 - 144):

“The net result of flapping, irrespective of the influence of tip vortices and wing taper, is a general increase in the force per unit area⁵² and inertial moment; a distal section of wing will experience greater aerodynamic force and greater inertial moment per unit area than a

comparable section more proximal on the wing (see Supplementary Appendix 1 for more details). In turn, this means that a change in morphology in distal wing sections will yield a greater relative change in the forces experienced there than would result from a similar, more proximal, change in morphology."

How steep does a gradient have to be to show evidence for mechanical sensitivity of wing shape?

Response: We have no *a priori* expectations of a threshold that would constitute a "significant" gradient versus a "non-significant" one. In fact, given that our predictions are based on physics of flapping and not all birds rely on flapping to the same degree (e.g. hummingbirds vs. condors), we expect that the steepness of the gradient might vary considerably among taxonomic groups. This is a topic that may prove interesting for subsequent studies (potentially with deeper sampling within flapping birds contrasted with specialized gliders), but is beyond the analyses that we are able to do with the present dataset. We have added a section to the Discussion exploring these thoughts (452 - 467):

"We predicted that evolutionary dynamics of wing morphology follow a gradient of mechanical sensitivity increasing from the base of the wing toward its tip. Based on simple aerodynamic and inertial models, we predicted that the general shape of the gradient would resemble an exponential curve (see supplemental material for details). However, we caution that we have made no explicit predictions as to the magnitude of that increase, or what might constitute a "significant" gradient. Because different bird taxa rely on gliding and flapping flight to varying degrees²⁸, we expect that the sensitivity gradient itself might be a feature that varies similarly. For example, we predict that the sensitivity gradient strongest for hummingbirds, which rely solely on flapping⁷⁴⁻⁷⁷ and weakest for specialized gliders like albatrosses and condors⁷⁸⁻⁸¹, with other lineages falling somewhere on a spectrum between those end points. Our finding that the values of σ^2 and disparity differ between passerine and non-passerine birds reinforces our assertion that the sensitivity gradient might vary among taxa. Our intent was to assess whether the large representation of passerines in our sample biased us toward finding a gradient of evolutionary dynamics if passerines follow that pattern, but other birds do not, and not to test for differences in the magnitude of the gradient among taxa, so we hesitate to interpret those results further."

What determines if there is a cutoff or discontinuity?

Response: We used regression discontinuity analysis (RDA) to identify significant discontinuities. RDA assesses whether the coefficients of regression equations differ across an arbitrary point (in our case, the wrist joint). We have explored other options for identifying the cutoffs between putative wing modules (see our responses to the detailed comments/questions below).

Broadly, there should be much much more description of the methods either in the text or in SI. As written, it would not be possible to replicate the methods, as you will see by the number of questions and clarifications in the methods section. By the time I reached the results, I was totally at a loss about how the authors carried out the CR analysis.

In the end, the authors might be correct in their conclusions, but it not currently possible to determine from the information presented.

Response: We have expanded our presentation of the methodology (see revised Methods section and detailed responses below). In particular, we have clarified that we did not take a geometric morphometrics approach as the core of our analyses, and instead focused on wing attributes that are expected from first-principles to directly influence the production of aerodynamic forces and affect the inertial moment of the wing during flapping flight.

Specific comments

Introduction

- Lines 63-66 are a little confusing in the way that "function" is used three times with seemingly different connotations.

Response: Upon revisiting this section of the manuscript, we agree and have revised it to streamline the wording (Lines 58 - 65):

“The evolution of individual morphological traits is not always directly and solely linked to their biomechanical function, though. Individual traits within a biomechanical system can contribute in varying degrees to the functional output of the whole. Traits working together to perform some function are the hallmark of morphological integration⁵⁻¹⁰. “Function” in this context can refer to the physical output of force to accomplish a variety of tasks including (but not limited to) feeding, locomotion, sexual display or competition, as well as a variety of other behavioral and potentially physiological attributes.”

- Hanging a lot of intellectual framework on the Felice et al paper. Is each link really independent (lines 74-75)? Can the handwing really exist without the armwing? Are they actually functionally separable, given that each is involved in the production of aerodynamic forces? I have similar questions about the Felice et al. argument.

Response: We thank the referee for this comment which pointed out to us that we 1) needed to improve our introduction of the concept of modularity, and 2) that we needed a more thorough treatment of the literature in that introduction. We would point out, though, that neither we nor Felice et al. argue for complete functional separation among putative modules – indeed, we

(and Felice et al.) define modules as “semi-independent”. Obviously the “whole” doesn’t function without the parts, either in 4-bar linkages or here with wings. However, there is still strong regionalization of both form and function in the 4-bars with documented evolutionary consequences, so it’s reasonable to have predicted that we might see similar modularity in wings.

With respect to functional separation of aerodynamics, flaps and ailerons of airplane wings are both responsible for producing aerodynamics forces... but to different effect – enhancing lift production in the wing vs. maneuvering. While there is some inherent folly in pointing to engineered wings as a vignette to support our point, it is not unreasonable to think that morphological adaptation in the AW might be more similar to the flaps, while adaptation in the HW might be similar to ailerons, albeit probably with less-clear distinction than the engineered solutions.

We have expanded our introduction to the concepts of morphological, functional, and evolutionary modularity, with particular emphasis on how morphological modules (*sensu* Olson and Miller, Cheverud, and others) are related to locomotor modules (*sensu* Gatesy and Dial), and how these ideas tie into evolutionary dynamics (Lines 58 - 77):

“The evolution of individual morphological traits is not always directly and solely linked to their biomechanical function, though. Individual traits within a biomechanical system can contribute in varying degrees to the functional output of the whole. Traits working together to perform some function are the hallmark of morphological integration⁵⁻¹⁰. “Function” in this context can refer to the physical output of force to accomplish a variety of tasks including (but not limited to) feeding, locomotion, sexual display or competition, as well as a variety of other behavioral and potentially physiological attributes.

Modularity exists when function differs between clusters of integrated traits⁵⁻¹⁰. The degree of integration among these coevolving traits and their organization into mosaics of semi-independent modules are mediated by their shared development and by the magnitude of their impact on functional output^{5-8,11-13}. Locomotor modules¹⁴ are a subset of functional modularity wherein regionalization of biomechanical function (i.e., running vs. flying) leads to the development of morphological (and potentially physiological) traits specific to those tasks. As natural selection acts to shape the functional output of biomechanical systems, each of the individual traits may experience selective pressure commensurate with their relative contribution to the output of the system^{10,11,13,15}. Therefore, in a biomechanical context, the strength of the relationships between morphological traits and their mechanical function (termed mechanical sensitivity) may be an important driver of their evolutionary dynamics (i.e. tempo and mode¹⁶⁻¹⁸).”

- Wings aren't 2d -- agreed. But wings are also not static in 3d. The authors certainly know this, but it represents an important caveat to this work overall. What looks morphologically similar in

3d (or 2d) might move dissimilarly in the dynamics of flight (and vice versa).

Response: This is certainly one of the caveats of our study, and we devote a section of our discussion to it (see Lines 480 - 494). We recognize, however, from the referee's comment that some foreshadowing is necessary in the introduction. To that end, we have added the following (Lines 110 - 112 and 150 - 152, respectively):

"Bird wings are also not static structures. The shape -both 2D and 3D- changes throughout the wingbeat cycle and is modified by birds, termed wing morphing, to accomplish various flight tasks⁴⁵⁻⁵²."

"Though wing shape changes dynamically through flight, the range of shapes that a wing can achieve via wing morphing is constrained by its static form. We investigated the relationship between mechanical sensitivity and the evolution of static wing morphology."

- Line 115, I think that the Gatesy & Dial citation is not appropriate here. Those authors were talking about locomotor modules not in the GM "modules" sense, but rather as evolutionarily decoupled complexes (forelimb, hind limb, tail -- not within wing functional separation).

Response: We did, indeed, intend to point to potential evolutionary decoupling between the HW and AW. In response to this question, we have clarified the presentation of our thinking about the links between morphological modules, locomotor modules, and how they are influenced by evolutionary pressures in the Introduction (please see earlier response). We also worked to clarify our intent in the section the referee pointed to in this comment (Lines 120 - 125):

"The two wing regions (HW and AW) may be under differential selective pressures, or subject to different selective or developmental tradeoffs and constraints leading to regionalization of biomechanical function *sensu*¹⁴ leading to evolutionary or morphological modularity within the wing^{12,53}. For these reasons, we hypothesized that morphological modularity exists in the wing, dividing it into discrete armwing and handwing modules."

Olson, E. C. & Miller, R. L. *Morphological Integration*. (University of Chicago Press, 1958).

Cheverud, J. M. Phenotypic, Genetic, and Environmental Morphological Integration in the Cranium. *Evolution* **36**, 499–516 (1982).

Cheverud, J. M. Developmental Integration and the Evolution of Pleiotropy. *American Zoologist* **36**, 44–50 (1996).

Klingenberg, C. P. Morphological Integration and Developmental Modularity. *Annual Review of Ecology, Evolution, and Systematics* **39**, 115–132 (2008).

Armbruster, W. S., Pélabon, C., Bolstad, G. H. & Hansen, T. F. Integrated phenotypes: understanding trait covariation in plants and animals. *Philosophical Transactions of the Royal Society B: Biological Sciences* **369**, 20130245 (2014).

Zelditch, M. L. & Goswami, A. What does modularity mean? *Evolution & Development* **23**, 377–403 (2021).

Methods

- Details of the PCA are not at all clear from the text (lines 165-174). It seems like the authors are doing a geometric morphometric analysis, but the usual discussion of those methods is missing (centering, scaling, etc.). Is this GM analysis? What steps did the authors use? What software, etc.? In GM, care must be taken to make sure that landmarks are homologous to one another or that there are sufficiently dense sliding or semi-sliding landmarks that homology can be approximated (it seems like this would be necessary for the CR analysis). Do the authors test whether upsampling large specimens or downsampling small specimens had an effect on the analysis? I would be concerned that different numbers of points per specimen might cause problems (or maybe this just reflects not understanding the approach).

Response: We did not take a geometric morphometrics approach in our analyses, and PCA was used only to automatically align the wings (see next response). Thus, the issues of landmark placement, centering, scaling, etc. are not relevant here. Instead, we directly measured variables that are known to influence the aerodynamics of wings.

We have clarified this in the text (Lines 201 - 205):

“We measured four shape traits from each wing slice, focusing on attributes of the wing that are expected from first principles to directly influence the aerodynamic forces and inertial moment of the wing. We favored this approach over a geometric morphometric (GM) approach because, while GM may provide higher resolution shape information, the link between form and function in a GM framework is less direct.”

The number of wing sections from which we took measurements was kept consistent across all taxa to facilitate direct comparison, section-by-section, rather than keeping dimensional section width constant and adjusting the number of sections to fit the size of the wing. Our initial version of the manuscript erroneously stated that we allowed the number of AW slices to vary, but we did not – there was a consistent 10 AW slices. We apologize for this error, and have corrected it (Lines 196 - 200):

“To standardize subsequent analyses, we restricted the dataset to 35 span-wise slices (25 HW and 10 AW), which reflects the number of slices in the taxon with the shortest AW. In addition to standardizing the analyses, substantial trauma occurs during the removal of the wing during

preservation, so excluding the proximal AW slices also reduces the influence of preservational artifacts.”

We investigated the effect of the number of sections in our analysis when first developing our methods. Because the sections are samples of a continuous curve, our analysis was not particularly sensitive to section count although very small sections become noisy due to serrations in the curve at feather overlaps, and large sections lose resolution around the wrist (see Figure 2 in the main text).

- Lines 171-172, Are the PC axes descriptions of the loadings? It is very unclear how these axes were determined.

Response: We only used the PC analysis as a means to automatically align the wing scans to a common coordinate space. We did this because the X, Y, and Z axes output by the scanner were somewhat haphazard – depending on the position and orientation of the wing during scanning. The PCA functionally “rotated” the vertices in the wing scans such that the long axis (wing length) is PC1 (and thus, the X axis for subsequent analyses), the second longest axis (chord) is PC2 (and the Y axis), and the thickness of the wing is PC3 (the Z axis). As such, the usual PCA outputs (loadings, etc.) are uninteresting. To avoid confusing readers, we have reduced our discussion of the PCA in Methods so that it does not seem as though it was central to our analyses (as it would be in a GM approach; Lines 186 - 188):

“We aligned the wing point clouds to a common coordinate system with the X axis extending along the length of the wing from base to tip, the Y axis along the chord from leading to trailing edge, and Z through the thickness.”

- Lines 175-178, how does this method alter the AW vs. HW dichotomy when there are differing numbers of segments in each (i.e., where the wrist falls relative to the segments)? Did the authors try a per-AW or per-HW approach?

Response: We did not analyze the handwing or the armwing separately, however we emphasize that none of the subsequent statistical tests are affected by having different numbers of slices between the handwing and the armwing. We would also point out that we kept the number of HW and AW slices (25 in the HW and 10 in the AW) consistent across taxa (though the initial version of the manuscript erroneously said otherwise).

- Can the authors be certain that using only 2/3 of the AW does not bias the results? Are any whole-bird scans available for validation?

Response: Unfortunately, we were not able to scan any whole-birds. The spread wing collections at the North Carolina museum is limited to wings that have already been removed from the body, and we were subject to the availability of their material, as we didn't not have permits to collect birds for this project. However, as both the aerodynamic and inertial effects are both

greatest at the wingtip, we do not expect that omitting the proximal portion of the wing seriously biased our finding. We have also taken steps to validate this assertion in response to specific questions raised by the second referee (please see our response below).

- Lines 192-193 -- using the species medians is probably not necessary. Phylogenetic comparative methods can utilize multiple observations per specimen. One of the Garland & Ives (or Ives & Garland) papers discusses the use of measurement error, which is statistically indistinguishable from having multiple specimens per species.

Response: We took this approach to facilitate inclusion of as many taxa in our study as possible. Many of the specimens in museum collections do not have direct measurements of their body mass (which we used for body-size correction), and we were therefore forced to use published median values for body mass in those taxa. For the sake of consistency, we decided to operate on species medians for all traits.

- Phylogenetic tree -- What was the source of the branch lengths and how were they scaled (if at all) when collapsing branches during pruning? As a phylogenetic tree is a hypothesis of relative relationships are there alternative phylogenetic hypotheses (topology or branch lengths)? How sensitive is the analysis to topology and/or branch length scaling?

Response: To assess the effect of tree topology (and thus, the underlying phylogenetic hypothesis) on our results, we iterated our evolutionary rate and morphological disparity analyses across 1000 posterior draws of the avian tree provided by Birdtree.org. We have added a figure to the supplemental material (supplemental Figure S2) with these results. The figure now depicts the median σ^2 across the wing, coupled with its confidence bounds. There was some variance around the median estimated σ^2 values, but the overall shape of the curve from the tip of the wing toward its base remained consistent. Therefore, our main result is robust to multiple phylogenetic hypothesis. Disparity, unlike σ^2 , is insensitive to tree topology.

We have detailed these analyses in the Methods (Lines 274 - 280):

“ A phylogenetic tree is a hypothesis of the relatedness among species. The analysis of σ^2 can be influenced by the assumed tree topology (branching structure and branch lengths). To assess the effect of tree topology on our estimates of σ^2 , we iterated our analysis across 1000 posterior draws of the bird tree with varying topologies. We then took the median σ^2 of the output at each wing slice along with the median absolute deviation (MAD) to summarize their central tendency and variance respectively (see supplemental material for details). Disparity, unlike σ^2 , is insensitive to tree topology.”

... and the Results (Lines 345 - 347):

“Finally, we also found that our results were robust to different phylogenetic hypotheses. The gradient pattern of σ^2 was replicated across 1000 posterior draws of the avian phylogeny (supplemental Figure S2).”

- I have some concerns about taxon sampling. Although passerines represent half of avian taxonomic diversity, morphologically they are rather homogenous. My gestalt feeling is that there is much more diversity in wing shape outside of passerines than within it. I wonder if some kind of rarefaction analysis could show that the authors are achieving "saturation" in wing disparity with the current sampling.

Response: Our sampling effort was somewhat, and unfortunately, curtailed by the pandemic. The museum collections were closed during lockdown and technicians were not available to scan wings. While we do not believe that a broader taxonomic sample would alter our findings, we certainly understand the referee's concern here. We took a multifaceted approach to investigate how our taxonomic sampling might have affected our inference. First, we conducted the suggested rarefaction analysis. Second, we analyzed only the passerines to see if they follow a similar trend to the overall result. Finally, we analyzed our dataset with the passerines removed. We find that the exponential decay pattern is consistent across all treatments. The rarefaction analysis showed that the variance in our σ^2 and disparity measurements increases as taxa are removed from the dataset (though not in a predictable way), but that the overall pattern of high σ^2 and disparity at the wingtip persisted. We have included a new figure in the supplement (Figure S4) that depicts the results of this analysis and detailed the analysis in Methods (Lines 262 - 273):

“ Passerines account for 113 of our 178 species taxonomic sample of birds. Though the Passeriformes is a large and morphologically diverse lineage, its strong representation in our sample could bias our main findings if the lineage differs systematically from other birds. We took two approaches to determine if evolutionary dynamics of wing morphology in passerines differ from other birds and thus influence the broader interpretation of our results. First, we divided our dataset into passerines and non-passerines, and applied the σ^2 and morphological disparity analyses as described above to each of those subsets. Second, we conducted a rarefaction analysis in which we removed taxa and iterated the σ^2 and disparity analyses on the remaining taxa. In this analysis, we randomly removed 1 to 172 (of a total 178) taxa, meaning that we conducted the σ^2 and disparity analyses on sets of 6 to 177 species; the removed taxa were replaced before subsequent subsampling. Further details of these analyses are presented in the supplementary material.”

... and Results (Lines 339 - 346):

“Effects of sampling and phylogenetic uncertainty

We found that the gradient patterns of both σ^2 and disparity were consistent across different partitions of the data (see supplementary Figures S3 and S4). Both passerine and non-passerine birds showed similar patterns of σ^2 and disparity, though the estimated values differed between the groups (see supplementary Figure S3). Also, a general gradient pattern is maintained through the rarefaction analysis (supplemental Figure S4), though random removal

of taxa affects the estimates of σ^2 and disparity. Finally, we also found that our results were robust to different phylogenetic hypotheses.”

- Lines 220-228 -- The authors need to include more information about the RDA analysis. What models were tested? How were they compared? It appears from Fig 1g that only linear discontinuity was tested (?vs. a linear model with no discontinuity?). This seems at odds with the theoretical prediction in Fig 1c of some kind of exponential decay in relative mechanical sensitivity (which seems like it might be an appropriate fit to the observed data). Also, maybe it is my right-handed bias, but I find it counterintuitive to have the axes run from tip to shoulder. I would have expected the reverse (which also might mean that a regular polynomial will fit well to these data).

Response:

The underlying first-principles model produces a non-linear curve which we expect would likely not show discontinuity when tested with a matching non-linear model. Nevertheless, the first principles model does predict a substantial change in the physical forces acting along the wing, with an inflection point near the wrist. Thus in the RDA we are testing whether the anatomical division between armwing and handwing is also a plausible discontinuity in the mechanical sensitivity model, not exhaustively searching for all possible inflection points. Please also see also CR results detailed in our response to the second reviewer where we detail our finding that a simple trend in the shape traits is not enough to create a signal of modularity.

We sympathize with the right-handed bias, but we made the decision to anchor the plots with the wingtip on the Y-axis as it was the most reliably and consistently identifiable point of our wing scans. The proximal-most slice of the armwing, on the other hand, represents a variable proportion of the armwing (most of it, in the case of the taxon with the shortest armwing, and maybe 1/3 of it in the taxa with the longest armwings – petrels, in our case). Because these details of slice enumeration might affect analysis, we prefer to draw some attention to them via the existing presentation scheme.

Results

- Lines 258-260 -- It's not clear how the CR analysis can be used on camber, chord, etc. The method was designed for multivariate shape data based on landmarks. Here it seems to be applied to scalar data. The fact that randomized data are not centered on CR = 1, suggests that something is amiss with this analysis.

Response: While it is certainly true that most applications, and indeed the vignettes provided by Adams in his introduction of the method, use geometric morphometrics input data. However, per Adams, 2015 (supplemental material), input may either be a 2D matrix of phenotypic trait

values or a 3D matrix of Procrustes-aligned coordinates, and Adams provided R code to implement both approaches. In this method, $CR = 1$ occurs when covariation is roughly equal among modules and within modules. It is our belief that randomizing the slice data actually creates a situation that is the inverse of a modularity signal: covariation between putative modules is greater than covariation within them. Hence, in this case, we recover $CR > 1.0$ from the randomization.

Reviewer #2 (Remarks to the Author):

The authors present a study on the modularity, mechanical sensitivity and evolution of wing shape across birds. Their main goal is to try and determine whether wing shape evolution (represented by disparity and evolutionary rates) is more driven by modularity in the wing, or mechanical sensitivity to wing loading and aerodynamics. In general I like this study a great deal and am happy to see some of these concepts applied to a new system. I only have one somewhat major critique and a few minor points for clarification.

Response: We thank the referee for the positive sentiment about our manuscript, and for the thoughtful commentary that followed.

My only “major” critique is with the modularity side of the analysis. I am not convinced that the modularity signal they find between the AW and HW sections isn’t actually just a methodological artefact. The CR method, if I have this right, essentially asks whether the trait values within modules are more close to each other than they are to trait values between modules. In this study, the traits are taken from a series of wing slices that run in series along the long axis of the wing. The two modules hypothesized in this study are also aligned next to each other in the same manner. It seems to be that it would be hard for these two to not be seen as separate modules. Unless the wing has a very strange shape, neighboring slices are always going to be more similar to each other relative to non-neighboring ones. Due to this, all the slices in HW, for example, are more close to each other, than the slices in AW with the exception for maybe the 1-2 near the boundary, but those are likely not enough to shift the overall pattern. In short, I believe the modularity signal being seen is due to comparing a series of neighboring features, which will be more similar to each other, against non-neighboring ones, making me question whether there really are two distinct modules here.

Response: We thank the reviewer for raising this concern because it is a central point in the interpretation of our modularity result. In a subtle distinction, the CR method asks whether covariation of trait values among taxa within a putative module is greater than the covariation across the modules – that is, it operates on the covariance matrices rather than on the raw trait values (see Adams 2015). This means that a simple trend in the trait data (linear or non-linear) that makes successive data points inherently similar to one another and less similar than more

distant points, like we see in the wing traits, is not actually sufficient to produce a modularity result. To verify this, we simulated wing trait data (with normal distributions) for 40 theoretical taxa at 35 “slices”, as in our empirical data set with six different underlying trends with and without discontinuities at the wrist (see Fig. S5). We then conducted the CR analysis on both the untransformed simulated data and on log-transformations of the data (as in our empirical analysis). None of the CR values were significantly different from 1.0 in our simulations.

We noted this in the manuscript with our presentation of the CR results (Lines 310 - 317):

“The wing shape traits that we measured showed increasing trends from the tip of the wing toward its base (see Fig. 2). The result of this is that the values of each shape trait for a given slice are inherently more similar to closely situated slices than they are to more distant ones. We tested whether the existence of an underlying trend in the shape data would bias us toward finding a signal of modularity using the CR method by simulating wing shape traits with no phylogenetic structure and measuring CR in the simulated data (see Supplemental Fig. S5). We found that the simple existence of a trend in the data, linear or otherwise, is insufficient to produce a signal of modularity.”

Some more minor clarifications:

Pg. 8, Ln 188-190: Does using the average mass for a bird species when the specimen mass is not available create a problem if the specimen in question ends up being an outlier in terms of size?

Response: It is certainly possible that an individual museum specimen could appear to have especially large wings in this scenario. This is part of why we used species medians for our morphological measurements and, where possible, used relatively large sample sizes ($n > 6$, in most cases). Our intent was to minimize the impact of outliers on our core analyses. Further, it’s unclear that an individual with over-inflated measurements like this would disrupt the modularity or evolutionary analyses.

Pg. 10, Ln 220-228: One aspect of the methods that I am fuzzy on is how some of these analyses are done when there are differing numbers of slices within the AW module. For instance, when looking at disparity of the trait values, it says that disparity is measured for each slice. But if there are not the same number of slices in the AW region across species, how do you determine which ones are used for a specific disparity measure? For instance, if one bird has 10 slices and another has 8, how are those 8 slices mapped onto the ten for measuring disparity? And do you have 8 measures of disparity or 10?

Response: This was an error in the original version of the manuscript, and we apologize for the misleading wording. In earlier iterations of our analysis, we attempted to allow the number of AW slices to vary so as to use all available data. This turned out to not work well for the reasons

identified by the referee above, so we changed our strategy to restrict the number of AW slices in all taxa to the number present in the taxon with the least AW slices. Unfortunately, the text was not updated prior to our initial submission to reflect the revised methodology. As we noted in a reply to similar concerns from the first referee, we have corrected the error in the text (Lines 196 - 200):

“To standardize subsequent analyses, we restricted the dataset to 35 span-wise slices (25 HW and 10 AW), which reflects the number of slices in the taxon with the shortest AW. In addition to standardizing the analyses, substantial trauma occurs during the removal of the wing during preservation, so excluding the proximal AW slices also reduces the influence of preservational artifacts.”

The discussion opens with several paragraphs which feel like a rehash of the introduction. Some of this can be streamlined/cut.

Response: We agree and have streamlined the beginning of the discussion (see revised Discussion).

Reviewers' Comments:

Reviewer #1:

Remarks to the Author:

I applaud the authors on their changes and improvements in this resubmission.

Reviewer #2:

Remarks to the Author:

The authors have addressed all of my original comments sufficiently.